# Antigen-specific Fab profiling achieves molecular-resolution analysis of human autoantibody repertoires in rheumatoid arthritis

Eva Maria Stork [1,4], Danique M. H. van Rijswijck [2,3,4], Karin A. van Schie [1], Max Hoek[2,3], Theresa Kissel [1], Hans Ulrich Scherer [1], Tom W. J. Huizinga [1], Albert J. R. Heck [2,3,5], Rene E. M. Toes [1,5] & Albert Bondt [2,3,5] ✉

The presence of autoantibodies is a defining feature of many autoimmune diseases. The number of unique autoantibody clones is conceivably limited by immune tolerance mechanisms, but unknown due to limitations of the currently applied technologies. Here, we introduce an autoantigen-specific liquid chromatography-mass spectrometry-based IgG1 Fab profiling approach using the anti-citrullinated protein antibody (ACPA) repertoire in rheumatoid arthritis (RA) as an example. We show that each patient harbors a unique and diverse ACPA IgG1 repertoire dominated by only a few antibody clones. In contrast to the total plasma IgG1 antibody repertoire, the ACPA IgG1 sub-repertoire is characterised by an expansion of antibodies that harbor one, two or even more Fab glycans, and different glycovariants of the same clone can be detected. Together, our data indicate that the autoantibody response in a prominent human autoimmune disease is complex, unique to each patient and dominated by a relatively low number of clones.

Antibodies play a central role in protecting the host from pathogens. To provide protection in a specific and tunable manner, each antibody harbors a highly variable region in its antigen-binding fragments (Fab). This variable domain is initially generated by recombination of variable (V), diversity (D) and joining (J) gene segments during which nucleotides are deleted, and palindromic and non-templated nucleotides are inserted at the junctions of the assembled gene segments[1–3]. Together with somatic hypermutation of the variable domain upon antigen encounter[4,5], the various processes may eventually give rise to billions of unique antibodies, including various clones capable of binding the same antigen[6]. This high diversity and flexibility of the antibody repertoire ultimately allows antigen-specific immune responses even against newly arising or continuously evolving pathogens.

To prevent the formation of antibodies that target the body itself, known as autoantibodies, various tolerance mechanisms are in place. These mechanisms identify autoreactive B cell clones and exclude them from the repertoire by, for instance, clonal deletion or editing of the variable domain[7]. Failure of tolerance mechanisms can ultimately lead to autoimmune diseases, currently estimated to affect about 1 in 10 individuals[8]. Notably, many autoimmune diseases are responsive to B cell-targeting therapies[9] and are accompanied by disease-specific autoantibodies[10].

Serological studies have provided insights into the association of autoantibodies with disease development, progression or treatment[10]. Consequently, autoantibodies are widely used as biomarkers for diagnosis and prognosis, for disease classification and for guiding

[1]Department of Rheumatology, Leiden University Medical Center, Albinusdreef 2, Leiden, The Netherlands. [2]Biomolecular Mass Spectrometry and Proteomics, Bijvoet Center for Biomolecular Research and Utrecht Institute for Pharmaceutical Sciences, University of Utrecht, Padualaan 8, Utrecht, The Netherlands. [3]Netherlands Proteomics Center, Padualaan 8, Utrecht, the Netherlands. [4]These authors contributed equally: Eva Maria Stork, Danique M. H. van Rijswijck. [5]These authors jointly supervised this work: Albert J. R. Heck, Rene E. M. Toes, Albert Bondt. ✉e-mail: a.bondt@uu.nl

treatment choices[10–12]. Yet, insights into the extent of autoantibody repertoires, i.e., the extent of tolerance failure, are lacking due to the limitations of the currently applied technologies.

We recently introduced a method that enables to study plasma antibody repertoires at the protein level with molecular resolution[13]. This liquid chromatography-mass spectrometry (LC-MS)-based Fab profiling approach selectively generates IgG1 Fab fragments from affinity enriched plasma IgG and analyzes these Fab molecules by LC-MS, thereby resolving the diversity of polyclonal antibody mixtures and repertoires based on the unique mass and retention time of each Fab molecule. The application of this approach revealed that plasma as well as virus-specific IgG1 repertoires are unique and polyclonal, with a few clones showing particularly high abundances[13,14]. This diversity of repertoires against infectious agents may, however, differ from that of autoreactive antibody repertoires as a result of the exclusion of autoreactive antibody clones by tolerance mechanisms as well as the different nature of and context in which autoantigens may be recognized. Here, we therefore introduce in-depth autoantigen-specific Fab profiling and resolve an autoreactive plasma antibody sub-repertoire at the molecular level, using the prominent autoantibody response of anti-citrullinated protein antibodies (ACPA) as an example.

ACPA target proteins containing citrulline, a post-translational modification of the amino acid arginine, and are highly specific to rheumatoid arthritis (RA), a systemic and highly prevalent autoimmune disease[8,15,16]. ACPA are detected in 50 to 75% of RA patients and associate with severe bone erosions, disease progression and poor treatment responses[17,18]. Moreover, ACPA can be present years before RA diagnosis[19] and frequently harbor N-linked glycans in their variable domains – also referred to as Fab glycans due to their localization in the Fab[20,21]. Intriguingly, the level of ACPA IgG Fab glycosylation in ACPA-positive healthy subjects correlates with transition to RA[22]. ACPA are thus not only a pivotal diagnostic and prognostic biomarker for RA, but also a promising candidate to predict RA development.

By adapting the total plasma IgG1 Fab profiling approach to the ACPA response, we here resolve the ACPA antibody sub-repertoire at molecular detail. We show that the ACPA repertoire in plasma is polyclonal, albeit dominated by only a few clones. We confirm that ACPA IgG1 are, relatively to total plasma IgG1, extensively Fab-glycosylated and, moreover, reveal that their repertoire is unique to each patient. Remarkably, the ACPA IgG1 repertoire is characterized by an expansion of antibodies harboring two or more Fab glycans, while at the same time a substantial fraction of ACPA IgG1 do not harbor any. Together, this study provides a means to characterize human autoantibody repertoires in-depth and unveils the complexity and unique features of a prominent human disease-specific autoantibody repertoire.

## Results

### ACPA-specific Fab repertoire profiling by LC-MS

To enable insights into the extent and molecular composition of autoreactive antibody repertoires, we introduce an in-depth autoantigen-specific Fab profiling approach using the prototypic autoantibody response in RA, the ACPA response, as an example. The procedure of ACPA IgG1 Fab profiling consists of three parts (Fig. 1a). First, ACPA are affinity purified from patient plasma using the cyclic citrullinated peptide 2 (CCP2), a "golden standard" antigen used in clinical practice for diagnostic and prognostic purposes, and its non-modified, arginine-containing version CArgP2; the specificity of the ACPA purification is monitored by enzyme-linked immunosorbent assays (ELISAs). Second, IgG is affinity captured from purified ACPA spiked with known concentrations of two monoclonal IgG1 antibodies; and ACPA IgG1 Fab fragments are generated by subsequent on-bead enzymatic digestion using the IgG1-specific protease immunoglobulin degrading enzyme (IgdE), which cleaves IgG1 just above the hinge region. Finally, the collected ACPA IgG1 Fab fragments are profiled

using the recently developed intact mass LC-MS-based method described in Bondt, et al.[13]; and the quantity and plasma concentrations of each Fab molecule detected are determined based on the monoclonal antibodies spiked.

To verify the approach, ACPA-positive RA patient plasma and ACPA-negative healthy donor plasma were subjected to the same procedure. As shown in Fig. 1b and c, ACPA IgG was successfully isolated from RA patient plasma, whereas tetanus toxoid (TT)-specific IgG (assessed as a control) was not detected anymore in purified ACPA. Subsequent LC-MS profiling of the intact Fab fragments generated from the captured ACPA IgG antibodies revealed a broad range of distinctive Fab molecules (Fig. 1d). Importantly, highly similar ACPA IgG1 Fab profiles were obtained when plasma from the same RA patient was applied multiple times, indicating the robustness of the approach (Fig. 1e and Supplementary Fig. 1). Of note, and in line with the absence of TT-specific IgG, only limited background levels of ACPA IgG reactivity and IgG1 Fab molecules were detected when ACPA-negative healthy donor plasma was analyzed (Fig. 1b to d).

To assess the impact of varying ACPA levels on the Fab profiles detected, we serially diluted an RA patient plasma with healthy donor plasma and applied the various mixtures to ACPA IgG1 Fab profiling. Known concentrations of monoclonal ACPA IgG1 and anti-TT IgG1 were spiked during ACPA purification as internal controls. As depicted in Supplementary Fig. 2a to d, serial dilutions resulted in a stepwise decrease in ACPA yield as well as in the total number and concentrations of ACPA IgG1 Fab molecules detected by LC-MS profiling. The contribution of the ten most abundant Fab molecules to the entire detected repertoire remained similar within a restricted range of dilutions (Supplementary Fig. 2c). Moreover, the monoclonal antibodies spiked were detected at about the expected concentrations (Supplementary Fig. 2e; ACPA IgG1 spiked at 2 μg/mL and detected in spiked plasma at 2.1 ± 0.6 μg/mL; anti-TT IgG1 spiked at 20 μg/mL and detected at 21.7 ± 4.3 μg/mL). In line with previous assessments of specificity, the monoclonal anti-TT IgG1 antibody was not detected in the ACPA IgG1 Fab profile, whereas the monoclonal ACPA IgG1 antibody was absent from profiles of ACPA-depleted plasma. Altogether, the approach of ACPA IgG1 Fab profiling thus showed to be highly specific, sensitive and robust.

### The ACPA IgG1 repertoire is diverse, unique to each patient, and dominated by a few molecules

Having the tool of ACPA IgG1 Fab profiling at hand, we applied the approach to plasma obtained from eight RA patients with varying ACPA IgG levels and ACPA fine-specificities (Supplementary Table 1, Supplementary Fig. 3). As depicted in Fig. 2a and Supplementary Fig. 4, all ACPA IgG1 repertoires consisted of hundreds of Fab molecules defined by their unique mass and retention time. Despite this large diversity of the ACPA IgG1 repertoire found within each patient, the repertoires of the various patients scarcely overlapped (Fig. 2b), and each repertoire showed a unique pattern (Fig. 2c, Supplementary Fig. 4), indicating that ACPA repertoires are unique per individual. Remarkably, the detected Fab molecules largely varied in their abundance (Supplementary Fig. 4), and each repertoire was dominated by only a few Fab molecules, with the ten most abundant Fab molecules making up ~29% (21–47%) of the entire detected repertoire (Fig. 2a, Supplementary Table 2). Patients 2, 7, and 8 are most striking in this respect as their repertoires were each dominated by only two molecules that were responsible for at least 15% of the ACPA IgG1 repertoire detected. Altogether, we conclude that each RA patient plasma comprises a unique ACPA IgG1 sub-repertoire which, albeit diverse, is dominated by a few molecules.

### ACPA IgG1 are distinctively Fab-glycosylated

To compare ACPA IgG1 with total plasma IgG1, we collected paired plasma IgG1 Fab profiles by applying plasma to the LC-MS-based Fab

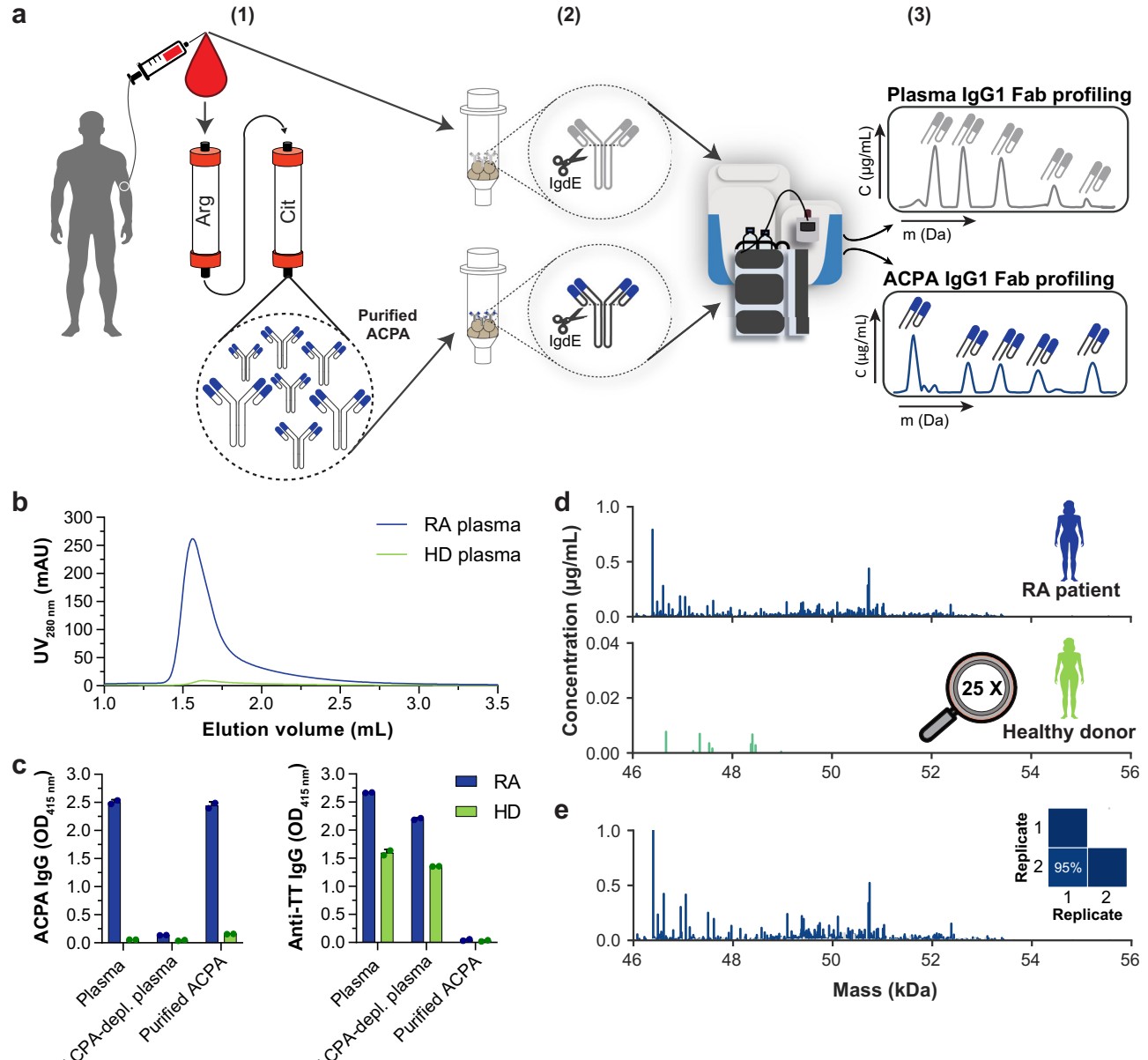

**Fig. 1 | Experimental workflow and validation of ACPA IgG1 Fab profiling.**
**a** Experimental workflow of the anti-citrullinated protein antibody (ACPA) IgG1 Fab profiling approach, which consists of three steps: (1) autoantigen-specific tandem affinity purification of ACPA from individual patient plasma using each one column functionalized with the non-specific CArgP2 control peptide (depicted as Arg) and the citrullinated CCP2 peptide (depicted as Cit); (2) affinity capturing of IgG from purified ACPA and subsequent generation of ACPA IgG1 Fab fragments by on-bead digestion using the IgG1-specific protease IgdE; (3) LC-MS-based Fab profiling of collected ACPA IgG1 Fab fragments. In comparison, total plasma IgG1 Fab profiles are obtained by applying total plasma directly to IgG affinity capturing (steps 2 and 3; top). **b**–**e** Validation of the approach by application of ACPA-positive rheumatoid arthritis (RA) patient plasma (blue) and ACPA-negative healthy donor (HD) plasma (green). **b** Chromatogram recorded following elution of ACPA affinity purification. **c** ACPA IgG and anti-TT IgG reactivity of plasma, ACPA-depleted plasma and purified ACPA determined by ELISA. All samples were assessed one time with technical replicates (n = 2) at the same dilution. Results are depicted as mean and standard deviation. Individual data points are overlaid. **d** ACPA IgG1 Fab profiles collected upon LC-MS-based Fab profiling. Unique Fab molecules are depicted as peak at their mass. **e** ACPA IgG1 Fab profile collected upon repetitive ACPA IgG1 Fab profiling. Profiling was repeated n = 2 additional times. The profile of one exemplary replicate is shown. Fab molecules shared between both replicates (blue) and unique for the profile depicted (grey) are indicated. The degree of overlap between both replicates is indicated as a heatmap.

profiling approach while omitting the initial ACPA purification (Fig. 1a). Notably, paired profiles overall poorly overlapped (Supplementary Fig. 5), and ACPA IgG1 Fab molecules were typically detected at distinctly lower concentrations (Supplementary Figs. 4 and 6). The concentration of ACPA IgG moreover rarely exceeded 1% of total IgG detected in the respective plasma (Supplementary Table 3). As shown in Fig. 3 and Supplementary Fig. 6, IgG1 Fab molecules principally exhibited masses between 46 and 49 kDa, which is in line with previous

datasets and the in silico generated distribution of Fab masses based on the ImMunoGeneTics information system (IMGT) database[13]. In sharp contrast, here detected ACPA IgG1 Fab molecules appeared at masses even above 54 kDa, and ACPA IgG1 Fab masses distributed unevenly across the mass range (Fig. 3, Supplementary Figs. 4 and 7).

Our previous research revealed that ACPA IgG harbor a markedly high extent of N-linked glycans attached to their variable domain, which evidently results in an increase in their mass compared to other

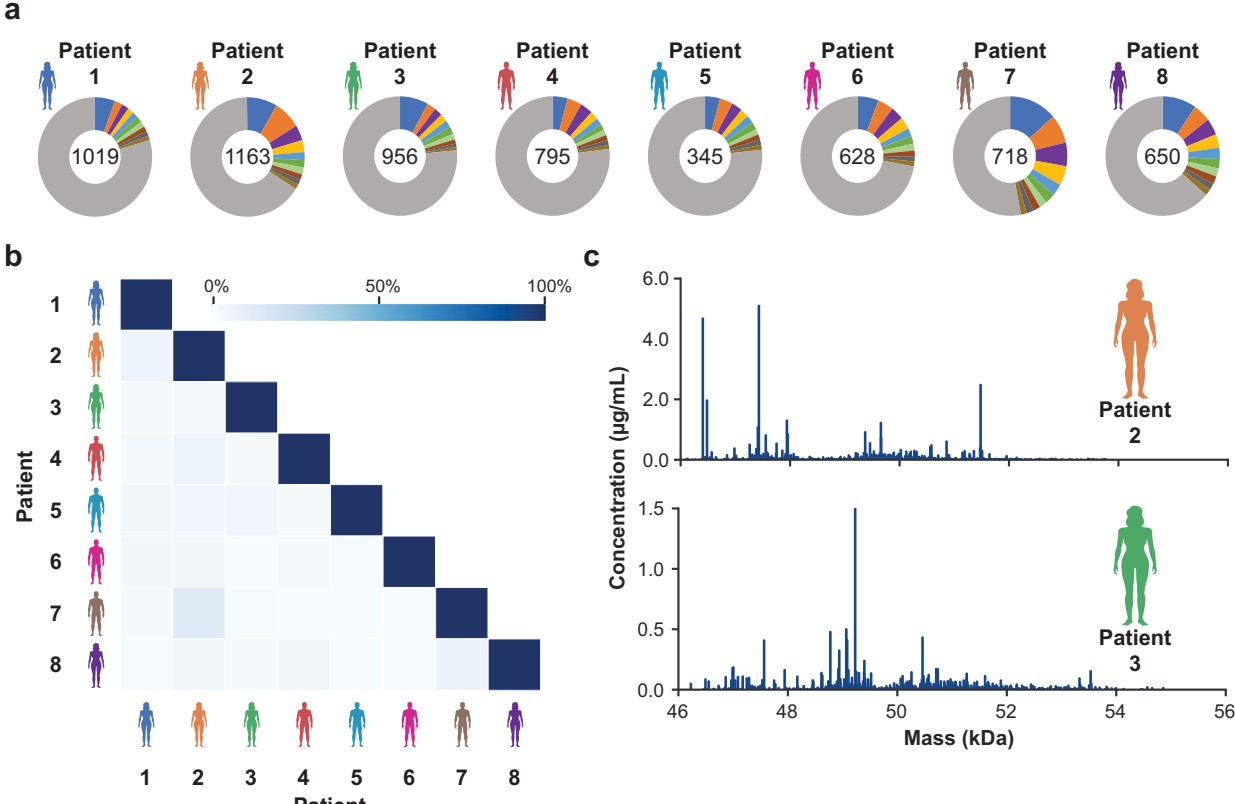

**Fig. 2 | Diversity, overlap and mass distribution of ACPA IgG1 repertoires determined for a cohort of RA patients (*n* = 8). a** Diversity of ACPA IgG1 repertoires observed within each patient and relative contribution of the ten most abundant Fab molecules to the total ACPA IgG1 repertoires detected. The total number of unique Fab molecules detected is depicted in the center of each donut plot. Each of the ten most abundant Fab molecules is indicated by a separate color; the remaining Fab molecules are indicated in grey. **b** Overlap between all ACPA IgG1 Fab profiles assessed. The degree of overlap is based on the number and abundance of overlapping Fab molecules and indicated by color gradient. **c** Exemplary ACPA IgG1 Fab profiles collected for two RA patients. Unique Fab molecules are depicted as peak at their mass. The height of the peak corresponds to the concentration of the respective Fab molecule.

IgG molecules[20,21]. Besides increasing the mass, protein N-glycosylation may introduce variation in glycoforms per protein. When profiling a monoclonal ACPA IgG1 that harbors N-glycans in its variable domain, we accordingly detected distinctive Fab molecules at different masses and retention times depending on the glycoform attached to the respective Fab fragment (Supplementary Fig. 8). Taking advantage of the mass and retention time changes detected for the various monoclonal antibody glycovariants, we therefore set out to seek for signature mass shifts within the ACPA IgG1 Fab profiles. Interestingly, our analysis revealed that numerous Fab molecules with increased masses could be linked to one or more glycovariants with distinctive signature mass shifts corresponding to galactosylation, sialylation or bisection (for an example, see Fig. 4a). Hence, the increased mass range observed for ACPA IgG1 Fab molecules can be explained by Fab glycosylation.

### ACPA IgG1 can harbor multiple Fab glycans

As delineated above, a sizeable fraction of ACPA IgG1 Fab molecules appeared to be glycosylated. Based on former ACPA IgG Fab glycosylation analysis[20,23–25], we estimated an average mass shift induced by a single ACPA Fab glycan to be ~2.4 kDa. A Fab fragment with one ACPA Fab glycan is thus expected at masses up to ~51.4 kDa. Here, we however observed a substantial number of ACPA IgG1 Fab molecules at even heavier masses (Fig. 4a, Supplementary Fig. 4). To further investigate this observation, we therefore defined mass ranges for Fab molecules harboring zero, one, or two or more glycans and quantified the relative proportion of these three categories in the patients' total plasma and ACPA IgG1 repertoires.

Based on these considerations, we extracted from our data that, on average, 61% (50–75%) of the ACPA IgG1 sub-repertoire is Fab-glycosylated, in contrast to only 6% (2–19%) of the total plasma IgG1 repertoire (Fig. 4b). While Fab-glycosylated total plasma IgG1 were almost exclusively detected in the mass range of Fab molecules that harbor just one glycan, all eight ACPA IgG1 sub-repertoires revealed a significant expansion of Fab molecules harboring two or more glycans, comprising 11% (5–15%) of the repertoire (Fig. 4c and d; *P* = 0.023). Our data thus confirms that ACPA IgG1 are overall extensively Fab-glycosylated but unveils that the presence and number of Fab glycans differs per molecule. Especially, a proportion of ACPA IgG1 does not harbor any Fab glycan and a relatively large number of ACPA IgG1 harbors two or more Fab glycans, a feature hardly observed in total plasma IgG1.

### Accounting for glyco-heterogeneity emphasizes dominance of a limited set of clones

The observation that most ACPA IgG1 are Fab-glycosylated also bears consequences for the estimation of clonal dominance and diversity within each repertoire: each of the glycosylated ACPA IgG1 Fab molecules detected may represent only one out of several glycovariants of the same ACPA IgG1 clone. For patient 7, for instance, six of the ten highest abundant Fab molecules could be matched as glycovariants based on the signature mass shifts determined above (Fig. 4a).

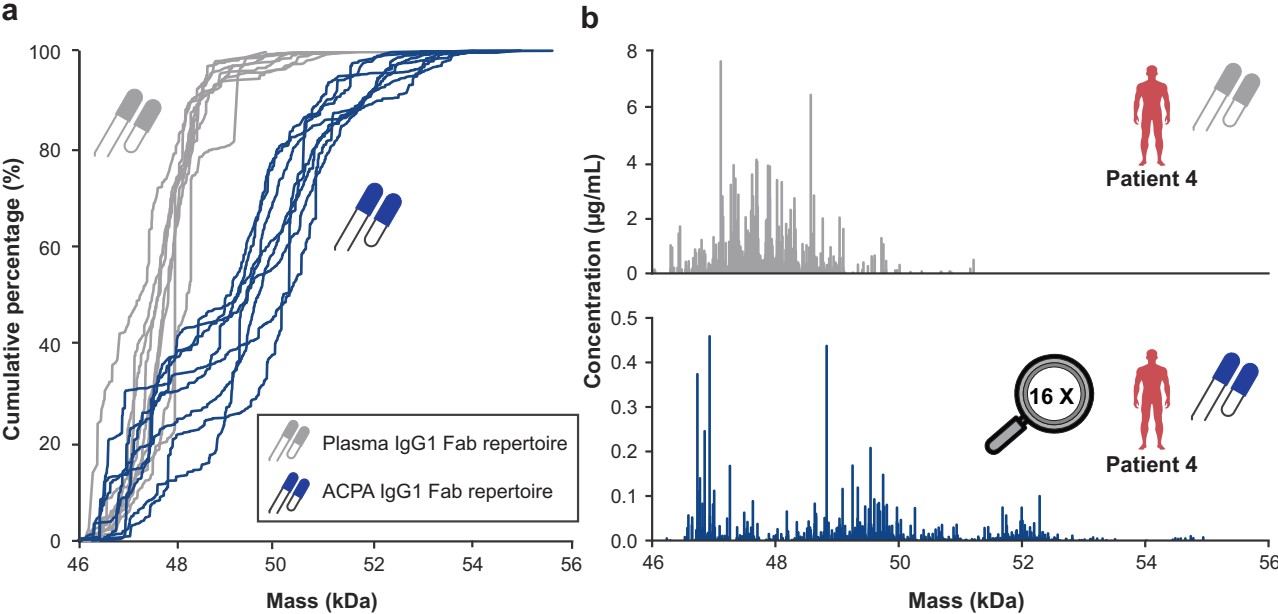

**Fig. 3 | Comparison of ACPA IgG1 versus total plasma IgG1 repertoires.**
**a** Cumulative percentage of total plasma IgG1 (grey) and ACPA IgG1 repertoires (blue) detected with increasing mass for $n = 8$ RA patients. Cumulative percentages are determined by summing the relative abundances of Fab molecules detected.

**b** Exemplary total plasma IgG1 (top, grey) and ACPA IgG1 (bottom, blue) Fab profile obtained for patient 4. Note: the 16-fold difference in the *y-axis* clearly reveals that ACPA IgG1 is a lower abundant sub-repertoire of total plasma IgG1.

Together with its various additional lower abundant glycovariants, this single clone covered more than 35% of the detected ACPA IgG1 repertoire. The clonal diversity of each ACPA IgG1 repertoire is thus lower than the molecular diversity determined by the sole Fab profiles (Fig. 2, Supplementary Table 2). Nevertheless, given that glyco-heterogeneity is concentrated in only three to six abundant glycovariants, ACPA IgG1 repertoires can be considered polyclonal; though they are each dominated by an even more restricted set of clones.

## Discussion

Autoreactive antibodies are hallmarks of many autoimmune diseases. Quantitative assessments of autoantibodies by serological testing served successful to unveil their fundamental associations with, for instance, disease development, progression or treatment and to establish autoantibodies as widely used biomarkers[10]. To understand the mechanisms underlying the development, maturation or dynamics of autoantibody responses, however, insights into the extent and molecular composition of autoantibody repertoires are required.

By B cell receptor sequencing of autoantigen-specific B cells, previous studies provided the first insights into the molecular composition of the autoreactive B cell compartment[26,27]. Yet, these studies focused on circulating memory B cells or the subset of antibody-secreting cells expressing their B cell receptor on the cell surface. As a consequence, antibody-secreting cells with low or no expression of B cell receptors on their cell surface and antibody-secreting plasma cells that reside, for instance, in the bone marrow or in inflamed tissues were not included. Particularly the antibody-secreting plasma cells, however, likely contribute substantially to autoantibody repertoires—especially in RA—as autoantibody levels change little upon therapeutic depletion of the CD20⁺ B cell compartment[28]. Moreover, B cell clones detected by B cell receptor sequencing are not necessarily found back as antibodies in circulation[29]. At present, the information on autoimmune repertoires at the antibody level are therefore limited, and novel approaches are needed to study autoantibody repertoires at molecular resolution in order to better comprehend the induction and evolution of autoantibody responses.

To this end, we here introduced an in-depth autoantigen-specific Fab profiling approach. After antigen-specific purification of the autoreactive antibody sub-repertoire, we generated IgG1 Fab fragments by selective digestion of IgG1 using the hinge-directed protease IgdE and separated the obtained Fab molecules based on their unique mass and retention time by means of intact mass LC-MS. Together, this approach allows to resolve the diversity of the autoreactive IgG1 repertoire. Spiked monoclonal antibodies of known concentration moreover enabled quantifying individual Fab molecules. Although differing variable domains may lead to Fab fragments with an identical mass and retention time and hence appear as the same Fab molecule, we expect such cases to be limited, given the low overlap between repertoires of different individuals observed in the current and previous studies[13,14]. Our validation of the approach further confirmed its specificity and sensitivity. Repetitive ACPA IgG1 Fab profiling of various aliquots of the same patient plasma furthermore yielded highly reproducible Fab profiles, and ACPA fine-specificity profiles were largely similar before and after ACPA purification, indicating that the observed Fab profiles are representative for the total circulating ACPA repertoire.

By applying this approach to plasma from eight RA patients, we here provide an in-depth characterization of a circulating autoantibody repertoire. Albeit previous work indicated the presence of independent antibodies confined in the ACPA repertoire[30], our characterization now unveiled its extent. Although part of the detected molecular diversity of each repertoire could be linked to glyco-heterogeneity, we conclude that the ACPA repertoire is polyclonal, consisting of at least dozens of unique ACPA clones. This is particularly interesting as the exclusion of autoreactive antibody clones by B cell tolerance mechanisms could conceivably result in a restricted set of autoreactive B cell clones. Whether the diversity of each ACPA repertoire results from a large number of initial B cell tolerance breaches or from diversification of initially breached B cell clones remains unknown due to the current lack of sequence information. Considering that ACPA-expressing memory B cells display a persistently activated, proliferative phenotype[31], it can, nevertheless, be speculated that continuous stimulation and activation of these cells leads to the

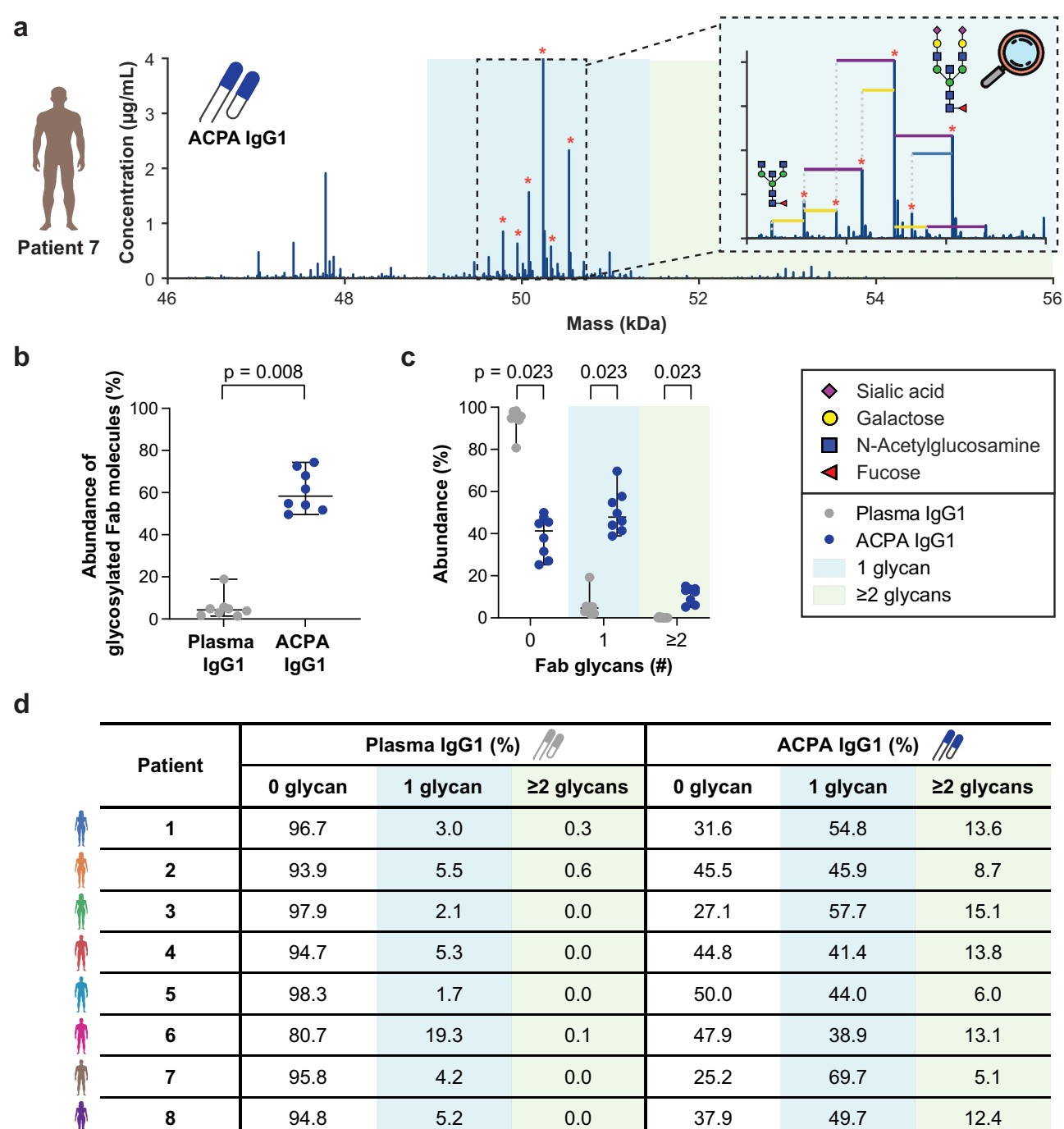

**Fig. 4 | ACPA IgG1 harbor high levels and variable numbers of Fab glycans, with an expansion of molecules harboring two or more Fab glycans. a** Exemplary ACPA IgG1 Fab profile, indicating the distribution of ACPA IgG1 Fab molecules across mass ranges of non-modified and mass ranges of glycosylated Fab molecules. Glycovariants detected for an exemplary high abundant ACPA IgG1 Fab molecule are indicated in the zoom. Differences of individual N-acetylglucosamines, galactoses or sialic acids are indicated as horizontal blue, yellow and purple lines, respectively. Glycovariants belonging to the ten most abundant Fab molecules are indicated by an asterisk. **b,c** Relative abundance of (**b**) glycosylated Fab molecules and (**c**) Fab molecules harboring zero, one, or two or more glycans among total plasma and ACPA IgG1 repertoires. Each dot

represents an individual RA patient. The median and range of all RA patients studied ($n = 8$) is indicated. P values calculated by two-tailed Wilcoxon matched-pairs rank test (c) and two-tailed Wilcoxon matched-pairs rank test corrected for multiple testing using the Bonferroni-Dunn method (d) are displayed. **d** Relative abundance of total plasma and ACPA IgG1 Fab molecules with zero, one, or two or more glycans. Relative abundances were determined based on the summed concentrations of all Fab molecules detected within the respective mass range and are indicated per RA patient. Mass ranges and relative abundances of Fab molecules with zero, one, or two or more glycans are highlighted by white, blue and green backgrounds, respectively.

diversification of the ACPA-B cell and thus, the eventually secreted ACPA repertoire.

Canales-Herrerias, et al.[32] previously reported that about 20% of IgG-secreting cells with the highest IgG production are responsible for about 50% of the total amount of secreted IgG. In line with this, the total plasma IgG1 repertoires of the RA patients studied were each dominated by only a few Fab molecules as were repertoires of total plasma IgG1 and IgA1 detected in our previous work[13,33]. Intriguingly, we here show that this dominance is not necessarily explained by single high IgG-producing cells of unknown specificities, but also holds true for an isolated autoantigen-specific sub-repertoire. Although the precise contribution of the most abundant Fab molecules may be affected by the depth of the respective profile as observed for ACPA IgG1 Fab profiling of a serially diluted RA patient plasma, the observed dominance of particular Fab molecules was persistent across different dilutions and, importantly, consistent across the eight studied RA patients. Notably, the dominance deduced from the sole Fab profiles is likely an underestimation since several glycosylated Fab molecules seemingly derived from the same antibody clone. Whether this dominance of unique ACPA molecules is caused by single ACPA-secreting cells with a high ACPA production rate or by an expansion of ACPA-secreting cells producing the same ACPA clone remains to be elucidated.

In line with previous observations indicating that up to 1% of IgG molecules is estimated to be an ACPA in an RA patient expressing ACPA at high levels[34], the ACPA IgG1 Fab molecules were hardly detected in total plasma IgG1 Fab profiles. Interestingly, ACPA IgG1 Fab molecules, however, mostly appeared at concentrations distinctly lower than the detected total plasma IgG1 Fab molecules. Consequently, the clones detected by total plasma IgG1 Fab profiling—albeit corresponding to a high percentage of the total IgG concentration—appear to be just the tip of the iceberg in terms of numbers, and the overall clonality of the total plasma IgG1 repertoire is presumably many times higher. With the aid of antigen-specific purification and the use of larger plasma volumes, ACPA IgG1 Fab profiling therefore not only allowed us to study an autoantigen-specific antibody repertoire at isolation; it also enabled us to visualize a sub-repertoire, which is mostly below the sensitivity limit of plasma IgG1 Fab profiling and thus not easily visible otherwise. Though, similar to plasma IgG1 Fab profiling, the detected ACPA IgG1 sub-repertoire may as well represent only a part of all ACPA IgG1, and the total diversity of each ACPA IgG1 repertoire may even be larger than revealed by our analysis.

In our previous work, we unveiled a special characteristic of ACPA IgG, namely the extensive level of glycans present in their variable domains. These variable domain glycans—also referred to as Fab glycans—result from the selective introduction of N-glycosylation sites during somatic hypermutation[27,35]. They may affect autoantigen binding[21,36], modulate the transport of the antibodies across membranes[37] and likely confer an advantage during B cell selection by regulating the threshold for B cell activation[36]. Since these glycans add additional mass to the protein backbone of the antibodies, ACPA IgG appear heavier as also shown by size exclusion chromatography and gel electrophoresis[21]. Using mass spectrometric analyses of the glycans released from isolated ACPA IgG, we previously estimated that, on average, more than 90% of ACPA IgG harbor such N-linked glycans in their variable domain[20,21].

Previous studies, however, only allowed estimation of the abundance of Fab glycans averaged over all ACPA IgG molecules. In contrast, the approach of ACPA IgG1 Fab profiling now enabled us to resolve the mass distribution of the contained molecules and hence to assess the presence and number of Fab glycans per secreted ACPA IgG1. Considering ~49 kDa as the maximum mass of a non-modified Fab molecule, we confirmed that the majority of ACPA IgG1 harbors Fab glycans and estimate the abundance of glycosylated Fab molecules in ACPA IgG1 repertoires to about 60%. Of note, this abundance of glycosylated Fab molecules is lower than previous estimations using released glycans. This is partly explained by our observation that a substantial fraction of ACPA IgG1 in each repertoire appears to carry two or more Fab glycans. Given a minimum mass of Fab fragments with two or more glycans of ~51.4 kDa, such molecules comprised on average 11% of the detected ACPA IgG1 repertoires, albeit the precise percentage varied between patients.

Even though not all ACPA IgG1 appeared to be Fab-glycosylated, including several of the most abundant Fab molecules, the observation that more than 10% of the ACPA IgG1 repertoires harbor two or more Fab glycans is interesting and clearly points to a particular enrichment. In comparison, only 0.3% of Fab sequences generated from the IMGT database possesses two N-glycosylation motifs (N-X-S/T, with X is not P), and no sequences harbor more than two. This notion also speaks from the analyses of total plasma IgG1 repertoires since for the assessed total plasma IgG1 repertoires at maximum 0.6% were estimated to carry two or more Fab glycans. Using the additional resolution provided by our method, we thus unveiled a significant expansion of ACPA IgG1 harboring multiple Fab glycans, which is not detected in total plasma IgG1, yet observed across RA patients. The reason for this enrichment remains unknown, but it would be interesting to investigate whether a similar enrichment is found for ACPA obtained from ACPA-positive pre-RA subjects, which might provide relevant insights into the evolution of the ACPA response across disease stages.

The power of ACPA IgG1 Fab profiling to resolve Fab glycosylation per ACPA molecule may, however, also come with limitations. Glycosylated antibody clones may be detected as several less abundant Fab molecules due to glyco-heterogeneity and are therefore more likely below the limit of detection. Moreover, we focus on the IgG1 subclass, the most abundant but not the only subclass of ACPA IgG[38,39]. Although fine-specificity profiles before and after ACPA purification were largely similar, some ACPA IgG1 clones may have been missed due to their lack of binding to CCP2. Lastly, our findings are consistent across all studied patients; yet biases due to the restricted number of individuals cannot be excluded.

Altogether, our approach sheds light on the extent of ACPA IgG1 repertoires and thereby provides detailed molecular insights into the autoantibody response underlying RA (Fig. 5). Given the important role of ACPA as a diagnostic and prognostic biomarker for RA[40], ACPA Fab glycosylation as a potential predictive marker for the transition from symptom-free autoimmunity to autoimmune disease[22] and the increasing awareness of Fab glycosylation also in other autoantibody species[41], the approach introduced here has high potential to further our understanding of autoantibody responses not only in RA, but also in autoantibody-characterized autoimmune diseases in general. Eventually, this contributes to study fundamental mechanisms underlying autoreactivity in human.

## Methods

### Study cohort

Patient plasma used in this study was collected from patients visiting the outpatient clinic of the Rheumatology Department at Leiden University Medical Center (LUMC) and selected based on ACPA levels. Healthy donor plasma was obtained from the biobank Rheumatic Diseases of the LUMC (protocol B14.011). All patients were diagnosed with ACPA-positive RA prior to sampling and met the 2010 American College of Rheumatology/European League against Rheumatism (ACR/EULAR) criteria for RA at the time of diagnosis. At the time of sampling, RA patients received different immunomodulatory treatments as listed in Supplementary Table 1. All donors gave written informed consent prior to sample acquisition. Permission for the conduct of the study was approved by the Ethical Review Board of the LUMC (protocols P13.171 and P17.151). Detailed patient information is provided in Supplementary Table 1.

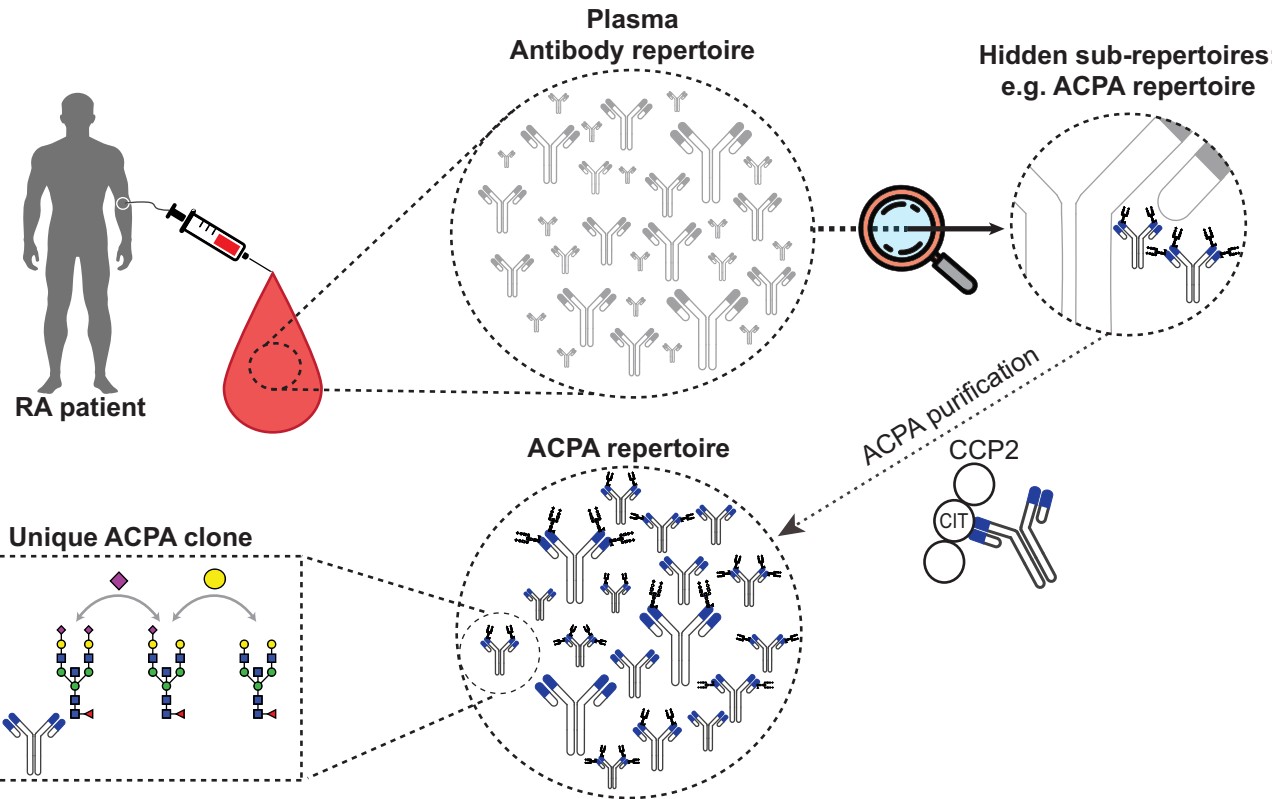

**Fig. 5 | In-depth molecular insights into the autoantibody response underlying RA.** ACPA IgG1 repertoires are a low-abundant sub-repertoire of total plasma IgG1 which can be revealed by antigen-specific ACPA IgG1 Fab profiling. The ACPA IgG1 repertoire is found to be diverse, unique to each patient and dominated by a few clones. Compared to plasma IgG1, ACPA IgG1 is extensively glycosylated with an expanded fraction harboring two or more glycans per Fab fragment. Moreover, different glycovariants of the same ACPA clone can be detected.

## Preparation of plasma

Plasma of patients 1, 2, 5 and 6 was collected from Ficoll-Paque-based isolation of peripheral blood mononuclear cells upon dilution of whole blood with phosphate buffered saline (PBS). Plasma of the remaining patients were collected upon centrifugation of whole blood for 10 minutes at 774 or 931 g and 20 °C. Plasma were stored at −20 °C until further use.

## Peptide synthesis, protein modification, and column preparation

For affinity-based ACPA purification, we selected the citrullinated antigen cyclic citrullinated peptide 2 (CCP2; patent number: EP2071335), a "golden standard" antigen used in clinical practice for diagnostic and prognostic purposes. The non-modified version of CCP2 (CArgP2) served to remove antibodies binding to either the column material or the peptide backbone.

ACPA fine-specificity profiles were determined using the citrullinated peptides fibrinogen α 27-43 (FLAEGGGVCitGPRVVERH), fibrinogen β 36-52 (NEEGFFSACitGHRPLDKK), cyclic citrullinated peptide 1 (CCP1; HQCHQESTCitGRSRGRCGRSGS), vimentin 59-74 (VYATCitS-SAVCitLCitSSVP) and enolase 5–20 (KIHACitEIFDSCitGNPTV)[30,42], as well as citrullinated fibrinogen and fetal calf serum (FCS). N-terminally biotinylated CCP2 and CArgP2 as well as the N-terminally biotinylated peptides to determine ACPA fine-specificity profiles were prepared (kindly provided by Dr. J. W. Drijfhout, Department of Immunology, LUMC) as described previously[42]. Proteins were modified as described before[43]; however, citrullination of FCS was performed at 37 °C and additionally supplemented with 2.5 M NaCl and 0.5 µM EDTA.

CCP2 and CArgP2 columns were prepared by functionalizing 1 mL HiTrap® Streptavidin HP columns (GE Healthcare/Cytiva) with 1.0–1.5 mg N-terminally biotinylated CCP2 or CArgP2 peptide at a flow rate of 0.2 mL/min. Excess of peptide was removed by washing with PBS and glycine-HCl, pH 2.5. Columns were stored in 20% ethanol until usage.

## Purification of ACPA from plasma

ACPA was purified from plasma using a tandem purification approach as depicted in Fig. 1a. In particular, CArgP2 and CCP2 columns were coupled in line to an ÄKTA pure purification system and equilibrated with PBS. Plasma was sterile filtered before respective volumes were manually injected into a 5 mL sample loop and applied to tandem purification at a flow rate of 0.5 mL/min. The unbound fraction was collected as ACPA-depleted plasma. After the sample application was completed, the sample loop and the columns were washed with PBS to remove the remaining unbound sample before ACPA was eluted from the CCP2 column with glycine-HCl, pH 2.5 at a flow rate of 0.5 mL/min. ACPA was collected in fractions of 0.2 mL. Immediately after fractionation, up to eight elution fractions starting from the beginning of the elution peak were pooled and desalted using 5 mL Zeba spin desalting columns (Thermo Fisher Scientific) according to the manufacturer's instructions. Desalted ACPA was stored at 4 °C until further assessment in enzyme-linked immunosorbent assays (ELISAs) and IgG capturing. CArgP2 and CCP2 columns were washed with glycine-HCl, pH 2.5 to ensure removal of remaining bound material and equilibrated with PBS for subsequent purifications. For consistency, ACPA was purified from 2 mL plasma for patients 1–7. For patient 8, 4 mL plasma was applied due to its markedly lower ACPA level.

## Enzyme-linked immunosorbent assays

ACPA IgG, tetanus toxoid (TT)-specific IgG and total IgG content of plasma, ACPA-depleted plasma and purified ACPA as well as ACPA fine-specificities were determined using in-house ELISAs. All samples were assessed with technical duplicates.

To determine ACPA IgG reactivity and levels, streptavidin-coated plates (Microcoat, standard capacity) were incubated for one hour at room temperature with 1 μg/mL N-terminally biotinylated CCP2 diluted in PBS/0.1% bovine serum albumin (BSA). Samples were diluted in PBS/1% BSA/0.05% Tween-20 (PBT) and applied for one hour at 37 °C. Bound ACPA IgG was detected by incubation with HRP-labelled rabbit anti-human IgG (DAKO, P0214; diluted 1:8000 in PBT) for one hour at 37 °C. ACPA fine-specificities were determined using the citrullinated peptides and proteins detailed above. Binding to citrullinated peptides was assessed by incubating streptavidin-coated plates with 10 μg/mL citrullinated peptide instead of 1 μg/mL CCP2. Binding to citrullinated proteins was assessed by coating Nunc Maxisorp plates (Thermo Fisher Scientific) with 10 μg/mL citrullinated protein diluted in coating buffer (0.1 M $Na_2CO_3$/0.1 M $NaHCO_3$, pH 9.6) overnight at 4 °C, followed by blocking with PBS/2% BSA for at least six hours. Samples were diluted in PBT and applied overnight at 4 °C before bound ACPA IgG was detected by incubation with the HRP-labelled rabbit anti-human IgG (diluted 1:3000 in PBT) for 3.5 hours at 4 °C. Unspecific reactivity to all peptides and proteins was determined by applying the non-modified instead of the citrullinated peptide or protein.

The presence of TT-specific IgG was assessed by coating Nunc Maxisorp plates with 1.5 LF/mL TT in coating buffer for three hours at 37 °C. Plates were blocked with 1% BSA/coating buffer for one hour at 37 °C before samples diluted in PBT were applied for two hours at 37 °C. Bound TT-specific IgG was detected by incubation with HRP-labelled rabbit anti-human IgG (DAKO, P0214; diluted 1:10,000 in PBT) for one hour at 37 °C.

To determine the total IgG content, Nunc Maxisorp plates were coated with 10 μg/mL goat anti-human IgG-Fc (Bethyl, A80-104) for one hour at room temperature, followed by blocking with PBS/1% BSA/50 mM TRIS, pH 8.0 for 30 minutes at room temperature. Subsequently, samples diluted in PBS/1% BSA/50 mM TRIS/0.05% Tween-20 (PBTT, pH 8.0) were applied for one hour at room temperature. Bound total IgG was detected by incubation with HRP-labelled goat anti-human IgG (Bethyl, A80-104P; diluted 1:20,000 in PBTT) for one hour at room temperature.

All ELISAs were read out using ABTS and $H_2O_2$ at an iMark Microplate Absorbance Reader (BioRad) or Multiskan FC Microplate Photometer (Thermo Fisher Scientific). ACPA IgG levels were quantified based on an in-house standard of pooled RA patient plasma, total IgG based on IgG standard serum (Merck Millipore, S1). All quantifications were performed using the Microplate manager software MPM-6 (BioRad).

## Monoclonal antibody production and purification

In-house produced monoclonal anti-TT IgG1 and ACPA IgG1 antibodies were expressed in Freestyle™ 293-F cells (Gibco) under glycoengineering conditions[42]. Briefly, cells were cultured in Freestyle™ 293 expression medium (Gibco) at 37 °C, 8% $CO_2$ on a shaking platform. Medium was supplemented with D-galactose substrate prior to transfection and cells transfected with plasmids encoding the immunoglobulin heavy chain, the immunoglobulin light chain, the large T antigen of the SV40 virus (GeneArt), the cell cycle inhibitors p21 and p27 (Invivogen), β1,4-N-acetylglucosaminyltransferase III (GnTIII), α2,6-sialyltransferase 1 (ST6GalT) and β1,4-galactosyltransferase 1 (B4GalT1) using 293-Fectin™ (Invitrogen). Transfection supernatants were harvested five to six days after transfection and purified using 1 mL HiTrap® Protein G HP affinity columns (GE Healthcare) according to the manufacturer's instructions. Eluted antibodies were re-buffered to PBS and single heavy and light chains excluded using a 53 mL HiPrep™ 26/10 Desalting column (GE Healthcare). Purified monoclonal antibodies were concentrated using Amicon® Ultra centrifugal filter units with a molecular weight limit of 50 kDa (Merck Millipore) according to the manufacturer's instructions. The purity and integrity of the monoclonal antibodies were assessed on SDS-PAGE.

An overview of the monoclonal antibodies used throughout this study, including their theoretical mass is provided in Supplementary Table 4.

## IgG capturing and subsequent IgG1 Fab generation

To capture IgG from purified ACPA and subsequently generate IgG1 Fab fragments, a similar procedure was used as described previously[13,14]. To this end, 20 μL CaptureSelect FcXL affinity matrix slurry (Thermo Fisher Scientific) was added to a Pierce Spin Column (Thermo Fisher Scientific) and washed three times with 150 mM phosphate buffer (PB). Purified ACPA was supplemented with 1% milk powder (Fantomalt, Nutricia) and 100 ng of an internal monoclonal antibody (mAb) standard (1:1 mixed trastuzumab: alemtuzumab). Purified and supplemented ACPA were incubated with the affinity matrix in fractions of up to 750 μL for each one hour head-over-head rotating at room temperature, and the unbound fraction was collected in between each incubation by centrifugation for one minute at 500 g. The affinity matrix with bound ACPA IgG was washed four times with PB after which ACPA IgG1 Fab fragments were generated by selective cleavage of captured IgG1 using the IgG1-specific protease immunoglobulin degrading enzyme (IgdE; branded FabALACTICA, Genovis), which cleaves human IgG1 just above the hinge region. For this purpose, the affinity matrix was resuspended in 50 μL PB containing 50 units of IgdE and incubated for at least 16 hours at 37 °C on a thermal shaker. The generated ACPA IgG1 Fab fragments were collected by centrifugation for one minute at 500 g.

To capture IgG from plasma, CaptureSelect FcXL affinity matrix was prepared in the same manner as described above. However, after washing, the affinity matrix was resuspended in 150 μL PB supplemented with 400 ng of internal mAb standard (1:1 mixed trastuzumab: alemtuzumab). A plasma volume containing an estimated 50 μg IgG1 was added and incubated with the affinity matrix for one hour shaking at room temperature. After four washes with PB, IgG1 Fab fragments were generated in the same manner as ACPA IgG1 Fab fragments.

## LC-MS-based Fab profiling

To examine the intact Fab fragments that were released, a reversed-phase liquid chromatography-coupled mass spectrometry (LC-MS) method and data processing was used, which was described previously[13,14]. In short, the collected intact Fab fragments were separated using a Vanquish Flex UHPLC instrument (Thermo Fisher Scientific) equipped with a 1×150 mm MAbPac Reversed Phase HPLC column (Thermo Fisher Scientific) and directly coupled to an Orbitrap Fusion Lumos Tribrid or an Orbitrap Exploris 480 MS with BioPharma option (Thermo Fisher Scientific). During chromatographic separation, both the column pre-heater and the analytical column chamber were heated to 80 °C. The Fab fragments were separated over a 62 minute gradient at a flow rate of 150 μL/minute. Gradient elution was achieved using two mobile phases, A (0.1% HCOOH in MilliQ water) and B (0.1% HCOOH in $CH_3CN$). At the start of the gradient, a mixture of 90% A and 10% B was used, ramping up from 10 to 25% B over one minute, from 25 to 40% B over 55 minutes, and from 40 to 95% B over the last one minute of the gradient. MS data were collected with the instrument operating in intact protein and low-pressure mode. Spray voltage was set at 3.5 kV, ion transfer tube temperature at 350 °C, vaporizer temperature at 100 °C, sheath gas flow at 15 arbitrary units, auxiliary gas flow at 5 arbitrary units and source induced dissociation (SID) at 15 V.

Spectra were recorded with a resolution setting of 7500 (at $m/z$ 200) in MS1. Scans were acquired in the range of 500–4000 $m/z$ using an automated gain control (AGC) target of 300% and a maximum injection time of 50 milliseconds. For each scan, 5 micro-scans were recorded. The raw spectra of the mass spectrometry data have been deposited to the MassIVE repository with identifier MSV000093196.

## Fab profiling data analysis

To analyze the LC-MS results, the retention time and mass (in Dalton) of all intact Fab molecules were retrieved from the generated RAW files using BioPharmaFinder (BPF) 3.2 (Thermo Fisher Scientific), similar to the method described before[13]. In short, deconvolution was performed using the ReSpect algorithm (Thermo Fisher Scientific) between 5 and 57 minutes using 0.1 minute sliding windows with 25% offset and a merge tolerance of 30 parts per million (ppm). Noise rejection was set at 95% and the output range between 10,000 and 100,000 Da with a target mass of 48,000 Da and a mass tolerance of 30 ppm. Charge states between 10 and 60 were included and the Intact Protein Peak model was selected.

Further data analysis was performed using in-house Python 3.9.13 scripts using libraries: Pandas 1.4.4[44], Numpy 1.21.5[45], Scipy 1.9.1[46], Matplotlib 3.5.2[47] and Seaborn 0.11.2. Masses of the BPF identifications were recalculated using an intensity weighted mean, considering only the most intense peaks comprising 90% of the total intensity. Masses between 45.0 and 56.2 kDa with the most intense charge state above 1000 $m/z$ and a BPF score ≥40 were considered to be Fab fragments of IgG1 antibodies. Fab fragments were considered to be identical within a defined mass and retention time window. The range of these windows was set to three times the standard deviation of the respective value observed for the internal mAb standard of the respective batch of samples that were processed with LC-MS. The windows were generally below 1.7 Da for the mass and below 1.4 minutes for the retention time. Concentrations of detected Fab molecules were determined by normalizing their sum intensity to the averaged sum intensity of the internal mAb standard and were corrected for the plasma volume applied to ACPA purification or total plasma IgG1 Fab profiling. The mAb standard was subtracted from the list of Fab fragments detected and the remaining Fab fragments were defined as unique Fab molecules based on their unique pair of mass and retention time. The degree of overlap between Fab profiles was determined by quantifying the relative abundance of overlapping Fab molecules and is shown as a percentage in the overlap heatmaps. Processed mass spectrometry data underlying donut plots, heatmaps, dot plots as well as raw data underlying depicted chromatography traces and the background-subtracted raw ELISA data are provided in the source data file.

## Coupling of glycovariants

To couple glycovariants of the same antibody clone, we took advantage of the mass and retention time shifts detected upon IgG1 Fab profiling of a monoclonal ACPA IgG1 antibody (7E4WT; for more information, see Kissel et al.[36]), harboring two N-linked glycans in its variable domain (Supplementary Fig. 8). Accordingly, glycoforms differing by a galactose were determined based on a mass shift of 162.1 Da and a retention time tolerance of 0.1 minute, those differing by an N-acetylglucosamine were determined based on a mass shift of 203.2 Da and a retention time tolerance of 0.1 minute, and those differing by a sialic acid were determined based on a mass shift of 291.3 Da and a retention time shift of less than 1 minute.

## Quantification of Fab molecules with zero, one, or two or more glycans

The proportion of Fab molecules harboring zero, one, or two or more glycans was defined using two key parameters: the median mass of Fab molecules in the ImMunoGeneTics information system (IMGT) database[48] (47.8 kDa) and the estimated average mass of a single ACPA IgG Fab glycan (~2.4 kDa). The latter was calculated based on the highly prevalent ACPA IgG Fab glycans G2FBS1, G2FS2 and G2FBS2, which are routinely used to determine ACPA IgG Fab glycosylation through glycan release[22–25]. Consequently, a Fab molecule with one glycan was estimated to have a mass of ~50.2 kDa, while the mass of a

Fab molecule with two glycans would be ~52.6 kDa, and the mass of a Fab molecule with three glycans would be ~54.9 kDa. We used these anticipated masses as the midpoint of each mass range and calculated the boundaries as follows: for Fab molecules with one glycan, the boundary was calculated as $50.2 \pm 1.2$ kDa (half the glycan); for Fab molecules with two or more glycans, the boundary was calculated as $52.6 - 1.2$ kDa and $54.9 + 1.2$ kDa. As a result, we established the following mass ranges: 46.0–49.0 kDa (Fab molecules without glycan), 49.0–51.4 kDa (Fab molecules with one glycan), and 51.4–56.1 kDa (Fab molecules with two or more glycans). The proportion of ACPA and plasma IgG1 repertoires harboring zero, one, or two or more Fab glycans was calculated as the relative abundance of Fab molecules within the respective mass range.

## Statistics

Statistical analysis of the extent and level of Fab glycosylation was performed using a two-tailed Wilcoxon matched-pairs rank test and corrected for multiple testing using the Bonferroni-Dunn method. Significant differences are defined as follows: not significant (ns; $P \geq 0.05$), *$P < 0.05$, **$P < 0.01$.

## Reporting summary

Further information on research design is available in the Nature Portfolio Reporting Summary linked to this article.

# Data availability

The raw spectra of the mass spectrometry data have been deposited to the MassIVE repository with identifier MSV000093196. Processed mass spectrometry data underlying donut plots, heatmaps, dot plots as well as raw data underlying depicted chromatography traces and the background-subtracted raw ELISA data are provided in the source data file. Source data are provided with this paper.

# Code availability

Custom code used for Fab profiling data analysis and generating Figures is adapted from Bondt, et al.[13] and van Rijswijck, et al.[14] and can be provided upon request.

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

## Acknowledgements

This research received funding through the Dutch Research Council (NWO) funding the Netherlands Proteomics Center through the X-omics Road Map program (project 184.034.019) as well as from the NWO gravitation program "Institute for Chemical Immunology" (Subgrant 00022, NWO-024.002.009 to T.W.J.H. and A.J.R.H.), ReumaNederland (LLP5 to T.W.J.H. and R.E.M.T.), the IMI funded project RTCure (777357 to T.W.J.H. and R.E.M.T.), Target to B! (LSHM18055-5GF to R.E.M.T.) and the European Research Council (ERC, GlycanSwitch, 101071386 to T.W.J.H.).

A.J.R.H. acknowledges support from NWO through the Spinoza Award SPI.2017.028. R.E.M.T. acknowledges support from the ERC as the recipient of an ERC advanced grant (AdG2019-884796). Views and opinions expressed are however those of the authors only and do not necessarily reflect those of the European Union or the European Research Council Executive Agency. Neither the European Union nor the granting authority can be held responsible for them. We moreover thank Gerrie Stoeken-Rijsbergen and Mirjam Damen for technical assistance, Dietmar Reusch and Markus Haberger (Roche, Penzberg) for the kind donation of trastuzumab, Genmab for the donation of alemtuzumab, as well as Jan Wouter Drijfhout for kindly providing the citrullinated and non-modified peptides.

## Author contributions

The performed research was conceptualized by K.A.v.S., T.W.J.H., A.J.R.H., R.E.M.T., and A.B.; patient material was collected by H.U.S. and T.W.J.H.; set-up of ACPA IgG1 Fab profiling, sample preparation and data analysis was performed by E.M.S., D.M.H.v.R., T.K. and A.B.; D.M.H.v.R. and M.H. developed the data analysis scripts; the manuscript was drafted by E.M.S., D.M.H.v.R., K.A.v.S., A.J.R.H., R.E.M.T. and A.B.; the final version was reviewed and approved by all authors.

## Competing interests

The authors declare no competing interests.
