## [Peer Review File · Nature Communications]

Antigen-specific Fab profiling achieves molecular resolution of human autoantibody repertoires in rheumatoid arthritisREVIEWER COMMENTS

Reviewer #1 (expert in autoantibody repertoire):

This very interesting paper is contributing to our understanding of the serum ACPA IgG repertoire and diversity. The data show that ACPA IgG1 antibodies are dominated by a limited number of antibody molecules, which could have implication for how we interpret RA serology.

Some specific questions:

1. How are the authors defining clones? How sensitive is the method in detecting close variants? Will the method discriminate clones with shared clonotypes within the same clonal tree?
2. If spiking several related monoclonal antibodies in the same sample, could they be detected and separated?
3. Did the patients display different ACPA fine-specificity profiles?
4. How was the ACPA fine-specificity maintained after purification?
5. Could the authors comment on why there were a lower absolute number of Fab clones detected when analyzing the total plasma IgG1 repertoire compared to the ACPA IgG1? What was the diversity index comparing total IgG1 and ACPA IgG1?
6. How is the ACPA IgG1 diversity comparing to diversity of other types of specific antibodies?
7. How large proportion of the total ACPA IgG is comprised of IgG1 compared to other subclasses?
8. Is the proportion of ACPA and the ACPA fine-specificity different when analyzing different IgG subclasses?
9. The overall level of Fab glycosylation, average 6% of IgG1 seems to be lower than what has been reported previously (15-25%) for total serum IgG using other methods (eg Bovenkamp PMID: 26851295). Is this due to the IgG1-specific analysis or is there another explanation?
10. The ACPA Fab glycosylation is interesting, but it would be good to highlight how this data is different from previously published data.

Reviewer #2 (expert in antibody technology):

This paper describes the resolution of circulating IgG1 in people diagnosed with rheumatoid arthritis using unique mass-spec methodology. It builds on prior work by the same team to map IgG1 following infection. Here, anti-citrullinated peptide antibodies are first purified, and then the ACPA IgG1 are resolved.

This does offer unique insight into ACPA, and the level of Fab-glycosylation is striking. The data/figures are comprehensive as shown.

Some comments for consideration:

1. The abstract is somewhat contradictory as written (and this is also reflected in other parts of the paper) - I'm not sure something can be "diverse" and also "dominated by a low number of clones"? Its also not always clear if the authors are referring to the inter-individual heterogeneity vs intra-individual diversity and clonal dominance, especially in the abstract. It would be helpful to use consistent descriptions for the major features highlighted within and between individuals throughout the paper and in the abstract.
2. The manuscript is focused on IgG1 only. It was hard to decipher why this was the case, I had to look back to previous papers from the same group to learn that the enzyme used to cleave/release IgG for mass-spec is IgG1 specific (IgdE). Could this be clearly explained in the

introduction/results instead?

3. While this approach gives very unique insight into IgG1, it doesn't enable investigation of IgG3. IgG3 is the other important subclass associated with autoimmune disease and inflammatory/immune mediated damage in autoimmune disease. I fully appreciate looking at IgG3 is beyond the current technology - given the specificity of the IgG enzyme and lower abundance of IgG3 in sera). But wonder if the authors might refer to this and other limitations, such as the small sample size, in their discussion?

Reviewer #3 (expert rheumatoid arthritis):

Stork and colleagues describe development of an autoantigen-specific liquid chromatography-mass spectrometry-based IgG1 Fab profiling approach using the prototypic anti-citrullinated protein antibody (ACPA) response in rheumatoid arthritis. The authors in-depth molecular characterization shows that each patient plasma harbors a unique ACPA IgG1 sub-repertoire which is diverse, albeit dominated by just a few antibody molecules. In contrast to the total plasma IgG1 antibody repertoire, the ACPA IgG1 sub-repertoire is characterized by an expansion of antibodies that harbor one, two or even more Fab glycans, and different glycovariants of the same clone can be detected. Together, the findings indicate that the autoantibody response in RA is complex, unique to each patient and dominated by a relatively low number of clones. These observations significantly advance our understanding of the ACPA response in RA, as well as of the nature of autoantibody responses mounted by individual RA patients.

This new approach is powerful for directly characterizing the antibodies present in the blood of patients with RA or with other immune responses of interest. It enables direct detection and analysis of antibodies present in blood or other biological fluids, and can be extended to other immune diseases and responses of interest. It also enables analysis of post-translational modifications of these antibodies, as the authors demonstrate in their analysis of Fab-glycosylation and other glycans.

One minor but important comment: Although the CCP2 peptides capture a subset of ACPA expressed in the vast majority of RA patients, several groups have shown that certain monoclonal ACPA from these same anti-CCP+ patients are non-reactive with CCP2. These non-CCP2-reactive ACPA are not captured and analyzed in the current study. Thus, what the authors are studying is the "CCP2-reactive antibody repertoire" in RA. The authors should address this limitation in the Discussion.

RESPONSE TO REVIEWERS' COMMENTS

with our responses in green

Reviewer #1 (expert in autoantibody repertoire):

This very interesting paper is contributing to our understanding of the serum ACPA IgG repertoire and diversity. The data show that ACPA IgG1 antibodies are dominated by a limited number of antibody molecules, which could have implication for how we interpret RA serology.

We highly appreciate the interest of the reviewer in the manuscript and its implications and thank the reviewer for the questions, which we address below.

Some specific questions:

1. How are the authors defining clones? How sensitive is the method in detecting close variants? Will the method discriminate clones with shared clonotypes within the same clonal tree?

We thank the reviewer for this question, as it is crucial for understanding the story. Fab molecules are defined by their unique combination of mass and retention time. Specifically, Fab fragments were considered identical within a defined mass and retention time window. The range of these windows is based on the variation observed for the monoclonal antibody standards that were spiked into each sample, and was set to three times the standard deviation of the respective value observed for these standards (see also lines 579 and following). Consequently, Fab fragments were deemed unique Fab molecules if their mass and retention time deviated more than three times the standard deviation observed for the standards, which was generally below 1.7 Da for the mass and below 1.4 minutes for the retention time. Thus, even single amino acid substitutions – for instance, of antibody clones within the same clonal tree – usually introduce a mass difference that is larger than the defined windows and can hence be detected as separate Fab molecules. Despite the great discriminatory power for unique Fab molecules, it remains beyond the scope of the approach to determine whether detected Fab molecules differ by single or multiple amino acids, i.e. whether they are clonally related or not. To enhance clarity, we revised the explanation how unique Fab molecules are defined in the methods section (lines 579 and following).

2. If spiking several related monoclonal antibodies in the same sample, could they be detected and separated?

If you were to spike multiple related monoclonal antibodies in the same sample, their detection and differentiation would be possible. As we explain in the response to question 1, even a slight variation in amino acid composition would lead to differences in mass, and possibly in retention time as well. Given that these monoclonal antibodies have mass and retention time differences larger than the defined windows, they can be identified and distinguished from one another.

3. Did the patients display different ACPA fine-specificity profiles?

We thank the reviewer for addressing this highly relevant topic. We did not assess the ACPA fine-specificities of the particular patient plasma before, but performed the required experiments following these questions. In line with what can be expected based on previous studies (Verpoort et al., Arthritis & Rheumatism, 2007; Woude et al., Annals of the Rheumatic Diseases, 2010), the eight patient plasma used in this study displayed varying ACPA fine-specificity profiles. See also our response to question 4.

4. How was the ACPA fine-specificity maintained after purification?

ACPA fine-specificities are generally maintained after purification due to the broad detection and capturing of ACPA enabled by CCP2 (Ioan-Facsinay et al., *Annals of the Rheumatic Diseases*, 2011; Pruijn et al., *Arthritis Research & Therapy*, 2010). Consequently, we found that the ACPA fine-specificity profiles before and after purification are largely similar. We appreciate the interest of the reviewer in this information and therefore extended Supplementary Figure 3 by the ACPA fine-specificity profiles before and after purification in response to this and the previous question. To further highlight the observations in the manuscript, we also added the following text: “we applied the approach to plasma obtained from eight RA patients with varying ACPA IgG levels and ACPA fine-specificities” (lines 158 to 159) and “ACPA fine-specificity profiles were largely similar before and after ACPA purification, indicating that the observed Fab profiles are representative for the total circulating ACPA repertoire.” (lines 304 to 306).

5. Could the authors comment on why there were a lower absolute number of Fab clones detected when analyzing the total plasma IgG1 repertoire compared to the ACPA IgG1? What was the diversity index comparing total IgG1 and ACPA IgG1?

Indeed, there was a lower absolute number of Fab molecules detected when analyzing the total plasma IgG1 repertoire in comparison to the ACPA IgG1 repertoire from the same patient. For technical reasons, however, the absolute number of detected Fab molecules is not the primary readout of Fab profiling. In addition to the technical properties of our approach, several factors contribute to this observation.

First, Fab glycosylation introduces another layer of complexity. The ACPA repertoire, being highly Fab-glycosylated, results in the distribution of the signal from a single glycosylated Fab clone across several detected Fab molecules as the result of glyco-heterogeneity. This dispersion occurs because these molecules, with the same protein backbone, differ in mass and potentially in retention time due to the varying glycan composition. Consequently, the same Fab clone can be detected as several different Fab molecules. For the ACPA repertoire, the number of detected ACPA IgG1 Fab molecules, e.g. displayed in Supplementary Table 2, is therefore an overestimation of the number of detected ACPA IgG1 Fab clones. In contrast, within the plasma IgG1 repertoire, most Fab molecules are not glycosylated, so that the vast majority of Fab molecules directly corresponds to a unique Fab clone without signal dispersion. To make this difference even more clear; we generally use the word “Fab molecules” throughout the manuscript which accordingly does not mean the same as unique “Fab clones”.

Second, the difference in sample volume may play a role. We initiated the analysis with ca. 5 to 25 μ L plasma to profile the plasma IgG1 repertoire. In contrast, a larger volume of 2 to 4 mL was used to profile the ACPA IgG1 repertoire. This increased volume used for ACPA IgG1 Fab profiling was necessary due to the low abundance of ACPA Fab molecules and our goal to profile the ACPA IgG1 repertoire in-depth. An increase in plasma volume used for Fab profiling typically coincides with an increasing sensitivity to detect low abundant Fab molecules as, for instance, shown in Supplementary Figure 2d. Consequently, the number of Fab molecules detected for the plasma IgG1 repertoire is an underestimation of the actual count, considering that there are likely Fab molecules below the limit of detection.

With regard to the diversity index, we focused on the relative dominance of the ten most abundant Fab molecules, and (considering the above mentioned reasons) to a lesser extent also species richness as represented by the total number of ACPA IgG1 and plasma IgG1 Fab molecules detected. Due to the above mentioned impact of differences in Fab glycosylation and profiling depth between ACPA IgG1 and total plasma IgG1 Fab profiling, we preferred not to directly compare the diversities of both repertoires within the manuscript. The respective section in the discussion is now refined for clarity (lines 341 to 348).

6. How is the ACPA IgG1 diversity comparing to diversity of other types of specific antibodies?

Insights into antigen-specific antibody repertoires with clonal resolution are so far restricted to this study and a previous study on IgG1 antibody repertoires enriched for binding to SARS-CoV2 spike protein (van Rijswijk, et al., Nature Communications, 2022). In both cases, a polyclonal response is evident. However, for ACPA IgG1, the diversity is even more pronounced, particularly when not accounting for Fab glycans. While coupling the different glycovariants helps mitigate this (see also our response to question 5 and the explanation in the results, lines 257 to 269), the ACPA repertoire remains notably diverse.

7. How large proportion of the total ACPA IgG is comprised of IgG1 compared to other subclasses?

Although the proportion of IgG1 varies between patients, IgG1 is typically observed as the most abundant subclass of ACPA IgG (e.g. Chapuy-Regaud et al., Clinical and Experimental Immunology, 2005; Lundström et al., PLOS ONE, 2014) and described to comprise more than 75% of the ACPA IgG repertoire (Lundström *et al.*, PLOS ONE, 2014).

8. Is the proportion of ACPA and the ACPA fine-specificity different when analyzing different IgG subclasses?

Similar to the proportion of IgG1, variation between patients is also observed for other subclasses. Yet, ACPA are described to be enriched for IgG3 (9% vs. 6% of total IgG) and IgG4 (6% vs. 2% of total IgG) at the cost of lower levels of IgG2 (7% vs. 17% of total IgG) (Lundström *et al.*, PLOS ONE, 2014). To the best of our knowledge, ACPA fine-specificities per different IgG subclass have not been analyzed so far. As studying additional subclasses and fine-specificities is beyond the scope of our work, we rather chose not to address this topic within the manuscript.

9. The overall level of Fab glycosylation, average 6% of IgG1 seems to be lower than what has been reported previously (15-25%) for total serum IgG using other methods (eg Bovenkamp PMID: 26851295). Is this due to the IgG1-specific analysis or is there another explanation?

We thank the reviewer for this question. Indeed, we also observed that the level of Fab glycosylation differs between different studies. This seeming discrepancy may have several reasons:

On the one hand, our analyses focuses indeed on the IgG1 subclass. Compared to IgG1, IgG4 was reported to show a strongly increased percentage of Fab glycosylation (43% and 44% vs. 12% for IgG1) (Koers et al., Journal of Allergy and Clinical Immunology, 2023; van de Bovenkamp et al., Proceedings of the National Academy of Sciences, 2018).

On the other hand, the level of Fab glycosylation also varies between donors. Patient 6, for instance, displayed a markedly increased level of Fab glycosylation compared to the remaining donors (19% vs. 6% on average). Such donor-specific variations were also apparent in our previous study on total plasma IgG1 Fab profiles (Bondt et al., Cell Systems, 2021).

Furthermore, detection at the level of intact Fabs may result in some underestimation of the level of Fab glycosylation. Fab profiling considers the intact Fab and resolves the various glycovariants of each individual Fab clone. The latter thereby goes along with the distribution of the signal intensity of glycosylated Fab clones across various Fab molecules as explained in the response to question 5. As a consequence, glycosylated Fab molecules may be detected less well than non-glycosylated Fab molecules.

Given the limited number of eight individuals in our study, we feel addressing the observed difference within the manuscript is beyond the scope. Instead, we extended the discussion to better indicate additional limitations of Fab profiling and our study that may contribute to this observation (e.g. lines 386 to 389 and lines 392 to 393: *“The power of ACPA IgG1 Fab profiling to resolve Fab glycosylation per ACPA molecule may, however, also come with limitations. Glycosylated antibody clones may be detected as several less abundant Fab molecules due to glyco-heterogeneity and are therefore more likely below the limit of detection. [...] Lastly, our findings are consistent across all studied patients; yet biases due to the restricted number of individuals cannot be excluded.”*).

10. The ACPA Fab glycosylation is interesting, but it would be good to highlight how this data is different from previously published data.

We thank the reviewer for the interest in ACPA Fab glycosylation and this indication. We consequently extended the paragraph in the discussion (lines 361 to 372):

“Previous studies, however, only allowed estimation of the abundance of Fab glycans averaged over all ACPA IgG molecules. In contrast, the approach of ACPA IgG1 Fab profiling now enabled us to resolve the mass distribution of the contained molecules, and hence to assess the presence and number of Fab glycans per secreted ACPA IgG1. Considering ~49 kDa as the maximum mass of a non-modified Fab molecule, we confirmed that the majority of ACPA IgG1 harbors Fab glycans and estimate the abundance of glycosylated Fab molecules in ACPA IgG1 repertoires to about 60%. Of note, this abundance of glycosylated Fab molecules is lower than previous estimations using released glycans. This is partly explained by our observation that a substantial fraction of ACPA IgG1 in each repertoire appears to carry two and more Fab glycans. Given a minimum mass of Fab fragments with two and more Fab glycans of ~51.4 kDa, such molecules comprised, on average, 11% of the detected ACPA IgG1 repertoires. Of note, the precise percentage varies between patients.”

To further highlight that our study considers the abundance of glycosylated Fab molecules, not Fab glycans, we changed the y-axis title of Figure 4b to “Abundance of glycosylated Fab molecules (%)”.

Reviewer #2 (expert in antibody technology):

This paper describes the resolution of circulating IgG1 in people diagnosed with rheumatoid arthritis using unique mass-spec methodology. It builds on prior work by the same team to map IgG1 following infection. Here, anti-citrullinated peptide antibodies are first purified, and then the ACPA IgG1 are resolved.

This does offer unique insight into ACPA, and the level of Fab-glycosylation is striking. The data/figures are comprehensive as shown.

Some comments for consideration:

1. The abstract is somewhat contradictory as written (and this is also reflected in other parts of the paper) - I'm not sure something can be "diverse" and also "dominated by a low number of clones"? Its also not always clear if the authors are referring to the inter-individual heterogeneity vs intra-individual diversity and clonal dominance, especially in the abstract. It would be helpful to use consistent descriptions for the major features highlighted within and between individuals throughout the paper and in the abstract.

We thank the reviewer for this comment and for highlighting the seemingly contradictory observations. We acknowledge that the ACPA IgG1 Fab profile of a particular patient can in fact show both, diversity and dominance of a low number of clones/molecules. The diversity thereby arises from the multitude of clones/molecules shaping the overall repertoire. Out of this multitude of detected clones/molecules, however, only a few stand out by their exceptionally high abundance and hence dominate the observed profile/repertoire. We appreciate the comment to clarify whether we refer to inter-individual heterogeneity vs. intra-individual diversity and clonal dominance and tried to clarify this aspect throughout the manuscript, including the abstract.

2. The manuscript is focused on IgG1 only. It was hard to decipher why this was the case, I had to look back to previous papers from the same group to learn that the enzyme used to cleave/release IgG for mass-spec is IgG1 specific (IgdE). Could this be clearly explained in the introduction/results instead?

We thank the reviewer for bringing this to our attention. The reviewer is correct that our focus is solely on IgG1, given our use of the IgdE enzyme, which specifically cleaves IgG1. We prefer to keep the explanation of IgG1 Fab profiling concise in the introduction, but revised the respective sentence in the results section (lines 103 to 106: "... Fab fragments are generated by subsequent on-bead enzymatic digestion using the IgG1-specific protease immunoglobulin degrading enzyme (IgdE), which cleaves IgG1 just above the hinge region.") and strengthened the IgG1 specificity of IgdE in the caption of Figure 1 (line 143), the discussion (line 294) and the respective sentence describing the procedure of IgG1 Fab generation in the methods section (lines 527 to 531).

3. While this approach gives very unique insight into IgG1, it doesnt enable investigation of IgG3. IgG3 is the other important subclass associated with autoimmune disease and inflammatory/immune mediated damage in autoimmune disease. I fully appreciate looking at IgG3 is beyond the current technology - given the specificity of the IgdE enzyme and lower abundance of IgG3 in sera). But wonder if the authors might refer to this and other limitations, such as the small sample size, in their discussion?

We appreciate the interest of the reviewer in profiling the IgG3 subclass and confirm that doing so is indeed beyond the current technology. In response to the question, we added a paragraph to the discussion referring to this and other limitations (lines 386 to 393):

“The power of ACPA IgG1 Fab profiling to resolve Fab glycosylation per ACPA molecule may, however, also come with limitations. Glycosylated antibody clones may be detected as several Fab molecules due to glyco-heterogeneity and are therefore more likely below the limit of detection. Moreover, we focus on the IgG1 subclass, the most abundant but not the only subclass of ACPA IgG^{38, 39}. Although fine-specificity profiles before and after ACPA purification were largely similar, some ACPA IgG1 clones may have been missed due to their lack of binding to CCP2. Lastly, our findings are consistent across all studied patients; yet biases due to the restricted number of individuals cannot be excluded.”

Reviewer #3 (expert rheumatoid arthritis):

Stork and colleagues describe development of an autoantigen-specific liquid chromatography-mass spectrometry-based IgG1 Fab profiling approach using the prototypic anti-citrullinated protein antibody (ACPA) response in rheumatoid arthritis. The authors in-depth molecular characterization shows that each patient plasma harbors a unique ACPA IgG1 sub-repertoire which is diverse, albeit dominated by just a few antibody molecules. In contrast to the total plasma IgG1 antibody repertoire, the ACPA IgG1 sub-repertoire is characterized by an expansion of antibodies that harbor one, two or even more Fab glycans, and different glycovariants of the same clone can be detected. Together, the findings indicate that the autoantibody response in RA is complex, unique to each patient and dominated by a relatively low number of clones. These observations significantly advance our understanding of the ACPA response in RA, as well as of the nature of autoantibody responses mounted by individual RA patients.

This new approach is powerful for directly characterizing the antibodies present in the blood of patients with RA or with other immune responses of interest. It enable direct detection and analysis of antibodies present blood or other biological fluids, and can be extended to other immune diseases and responses of interest. It also enables analysis of post-translational modifications of these antibodies, as the authors demonstrate in their analysis of Fab-glycosylation and other glycans.

One minor but important comment: Although the CCP2 peptides capture a subset of ACPA expressed in the vast majority of RA patients, several groups have shown that certain monoclonal ACPA from these same anti-CCP+ patients are non-reactive with CCP2. These non-CCP2-reactive ACPA are not captured and analyzed in the current study. Thus, what the authors are studying is the “CCP2-reactive antibody repertoire” in RA. The authors should address this limitation in the Discussion.

We thank the reviewer for this question. Overall, CCP2-based purification of ACPA enables co-purification of ACPA fine-specificities as it was previously shown by e.g. Ioan-Facsinay *et al.* (Ioan-Facsinay *et al.*, *Annals of the Rheumatic Diseases*, 2011). To visualize the effect of CCP2-based ACPA purification on the ACPA repertoire, we have now included the ACPA fine-specificity profile of each patient before and after ACPA purification (see Supplementary Figure 3). Although we find that the ACPA repertoire is largely similar before and after purification, we acknowledge that it cannot be excluded that some ACPA were missed due to their lack of binding to CCP2. We therefore added this limitation to the discussion (lines 390 to 392).

REVIEWERS' COMMENTS

Reviewer #1 (Remarks to the Author):

Thank you for the clear and extensive response and the revised manuscript. I have no further questions.

Reviewer #2 (Remarks to the Author):

The authors have addressed all my comments, and it's good to see clarity on the enzyme used (and specificity for IgG1) as well as a section on limitations in the discussion.

RESPONSE TO REVIEWERS' COMMENTS

with our responses in green

Reviewer #1 (Remarks to the Author):

Thank you for the clear and extensive response and the revised manuscript. I have no further questions.

We thank reviewer #1 for reviewing our manuscript and being satisfied by the comments.

Reviewer #2 (Remarks to the Author):

The authors have addressed all my comments, and it's good to see clarity on the enzyme used (and specificity for IgG1) as well as a section on limitations in the discussion.

We thank reviewer #2 for reviewing our manuscript and providing valuable feedback. We are pleased to hear that the clarity has been improved.